# Detection of Residual Film on the Field Surface Based on Faster R-CNN Multiscale Feature Fusion

Tong Zhou [1], Yongxin Jiang [2], Xuenong Wang [2], Jianhua Xie [1,*], Changyun Wang [1], Qian Shi [1] and Yi Zhang [1]

1   College of Mechanical and Electrical Engineering, Xinjiang Agricultural University, Urumqi 830052, China;
    3160502030@caa.edu.cn (T.Z.); 3180100326@caa.edu.cn (C.W.); 6180401111@caa.edu.cn (Q.S.);
    3180100339@caa.edu.cn (Y.Z.)
2   Research Institute of Agricultural Mechanization, Xinjiang Academy of Agricultural Sciences,
    Urumqi 830091, China; 20212109084@stu.shzu.edu.cn (Y.J.); zj34@xjie.edu.cn (X.W.)
*   Correspondence: 20212109027@stu.shzu.edu.cn

**Abstract:** After the residual film recycling machine recovers the film, some small pieces of the film will remain on the surface of the field. To solve the problem of collecting small pieces of film, it is necessary to develop a piece of intelligent picking equipment. The detection of small pieces of film is the first problem to be solved. This study proposes a method of an object detection algorithm fusing multi-scale features (MFFM Faster R-CNN) based on improved Faster R-CNN. Based on the Faster R-CNN model, the feature pyramid network is added to solve the problem of multiscale change of residual film. The convolution block attention module is introduced to enhance the feature extraction ability of the model. The Soft-NMS algorithm is used instead of the NMS algorithm to improve the detection accuracy of the model in the RPN network. The experimental results show that the model is able to effectively detect surface residual film in complex environments, with an *AP* of 83.45%, $F_{1\text{-score}}$ of 0.89, and average detection time of 248.36 ms. The model is compared with SSD and YOLOv5 under the same experimental environment and parameters, and it is found that the model not only ensures high-precision detection but also ensures real-time detection. This lays the theoretical foundation for the subsequent development of field surface residual film intelligent picking equipment.

**Keywords:** residual film; Faster R-CNN; multiscale features; attention module; intelligent picking equipment

## 1. Introduction

Plastic film mulching technology is widely used in the planting of more than 40 types of crops, such as grain, cotton, oil, vegetables, fruits, etc., due to its advantages in improving crop production and shortening the growth period of crops [1–3]. Currently, agricultural film is mainly composed of polyethylene, which has a long degradation cycle. If it is not recycled in a timely manner, it will not only pose a threat to the agricultural environment, but it will also affect the growth of crops and cause crop yield reduction [4–6]. Regarding an important high-quality cotton cultivation base in Xinjiang, China, the amount of residual film in the soil increases with the increase in mulching time and mulching area. This is due to the long-term adoption of mulching patterns, which consequently cause serious pollution to the farmland [7]. This study aims to develop an effective residual film recycling machine with an underlying background in the demands of intelligent agricultural equipment. There are broad application prospects to develop intelligent picking equipment for field surface residual film based on image recognition. Therefore, how to accurately identify and locate the residual film on the field surface is one of the core technologies for the development of intelligent picking equipment for residual film on field surfaces.

Satellite remote sensing technology comprises the characteristics of wide coverage and real-time updates, which are used by relevant scholars to detect situations concerning

farmland mulching [8,9]. Lu constructed a decision tree classifier by analyzing the rules of the spectral characteristics of the area of plastic-mulched landcover in Landsat 5 imagery to extract mulch information from images [10]. Hasi determined the most effective feature set and the optimal period for PMF mapping based on single-temporal and multi-temporal Landsat 8 imagery [11]. Xiong used classification and regression tree to analyze the characteristics of film-covered farmland in remote sensing images, constructed the rules of the film-covered farmland mapping algorithm, and tested it in Xinjiang, China [12]. Zheng used the random forest algorithm after optimizing key parameters to achieve a more accurate identification of the distribution of film-covered farmland, based on the Google Earth Engine cloud computing platform and Landsat 8 reflectance data [13]. Satellite remote sensing technology can effectively detect the film coverage of farmland. However, due to the high cost and low resolution of satellite remote sensing images, the detection of residual film debris on the fields is not ideal.

UAV low-altitude imaging technology has been widely used in crop monitoring, pest monitoring, and accurate identification of crop information due to its low cost, high image resolution, and strong operational flexibility [14–16]. In recent years, some scholars have applied UAV low-altitude imaging technology to farmland residual film identification. Liang collected field film images via drones and used different segmentation algorithms to identify them. The results showed that the iterative threshold segmentation algorithm had the highest target recognition rate, reaching 71% [17]. Wu proposed a pulse-coupled neural network algorithm based on the S color component, based on the UAV remote sensing visible light image, which has a good recognition effect on farmland residual film in different periods [18]. Zhu proposed a method for extracting mulched farmland, based on high-resolution RGB aerial images, that can connect mulches into blocks, and then obtained the distribution information of mulched farmland by area threshold segmentation [19]. Sun constructed a fully convolutional neural network by the multiscale fusion method, which can accurately obtain the geographical distribution and area of greenhouses and plastic film farmland, with an average overall accuracy of 97% [20].

The above-mentioned farmland plastic film recognition methods, based on satellite remote sensing technology and UAV low-altitude imaging technology, are aimed at farmland plastic film at the initial stage of plastic film coverage or without residual film recycling machine operation. At this time, the plastic films cover a large area and have good integrity; thus, the classification algorithm can be used to obtain recognition results with high accuracy. The films that remain on the field surface after the residual film recycling machine's operation, are no longer joined into strips and they become fragmented into different sizes and shapes. If the above-mentioned farmland plastic film recognition methods are used to identify the residual film, the accuracy rate will be reduced. To solve this problem, this study constructs a residual film detection model incorporating multiscale features, which can effectively improve the detection rate of small residual films on the field. The model can later be combined with film picking equipment. The model identifies the location of the residual film, and the control device controls the picking device to pick up the film, laying the research foundation for the development of an autonomous picking device. The main objectives of this study are:

(1) Construct a dataset of residual film images containing different light intensities.
(2) The Faster R-CNN model is improved by adding a feature pyramid network structure with an attention mechanism to the backbone feature extraction network. Additionally, instead of using the NMS algorithm, the Soft-NMS algorithm is used in the RPN structure.
(3) Test and verify the improved model and compare the performance of the improved model with other object detection models in the same experimental environment.
(4) Design and develop the residual film recognition software based on the improved model.

## 2. Materials and Methods

### 2.1. Dataset of Residual Film

Since there are no publicly available datasets of residual film images on the web, this study constructed a suitable dataset. For this study, in October 2022, a Sony camera (SONY ILCE-6400, Sony China Corporate Portal, Beijing, China) was used to photograph the leftover film on a field at Shihezi University's teaching experimental site in Shihezi, Xinjiang. To ensure the diversity of residual film images, the residual film was collected from the surface of the field under different light intensities. During the shooting process, the camera lens and the ground maintained a parallel position from the ground height of 80–110 cm. The captured images were uniformly cropped to 600 pixels × 600 pixels and saved in XX.JPG format. The blurred images were removed from the acquired residual film images by manual inspection, and 1000 residual film images were obtained, some of which are shown in Figure 1.

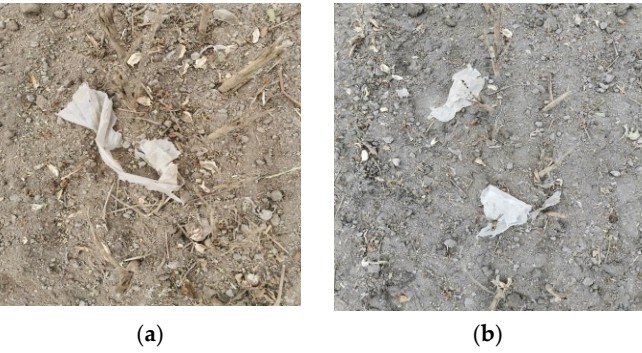

(**a**)　　　　　　　　　(**b**)

**Figure 1.** Residual film samples with different light intensities. (**a**) Strong light environment for shooting; and (**b**) low light environment for shooting.

Expansion of the image dataset was performed to avoid the phenomenon of overfitting during model training due to the small number of images in the dataset [21]. We randomly performed operations such as center cropping, rotating, and adjusting brightness for the acquired residual film images to expand the number of residual film images, and finally obtained a dataset containing 3000 images of residual film. The effect of image enhancement is shown in Figure 2. The 2100, 600, and 300 images in the residual film dataset were randomly selected as the training set, test set, and validation set, respectively.

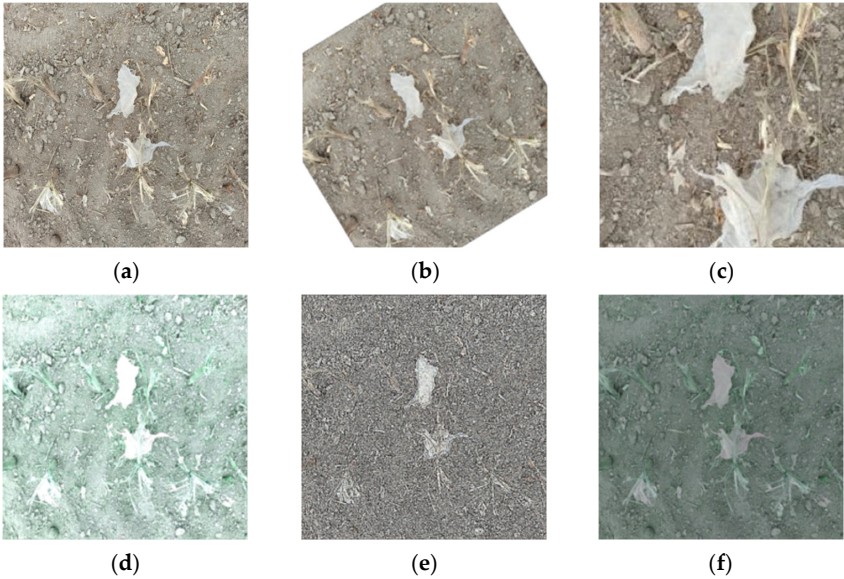

(**a**)　　　　　　　　(**b**)　　　　　　　　(**c**)

(**d**)　　　　　　　　(**e**)　　　　　　　　(**f**)

**Figure 2.** Effects of enhanced images. (**a**) Initial image; (**b**) rotation; (**c**) center cutting; (**d**) enhance brightness; (**e**) sharpen image; and (**f**) reduce brightness.

Image annotation is not only the key to constructing the residual film dataset but also a central step to ensuring the accuracy of residual film detection. In this study, we used LabelImg (Version No.1.8.6) to label the rectangular box of the residual film region in the expanded images, and the label of the residual film was set to residual plastic film to create a dataset in PASCAL VOC format, and an XML file was generated for each image. As shown in Figure 3, the XML file contains the file name, image name, image size, label name, and information about the location of the residual film in the image.

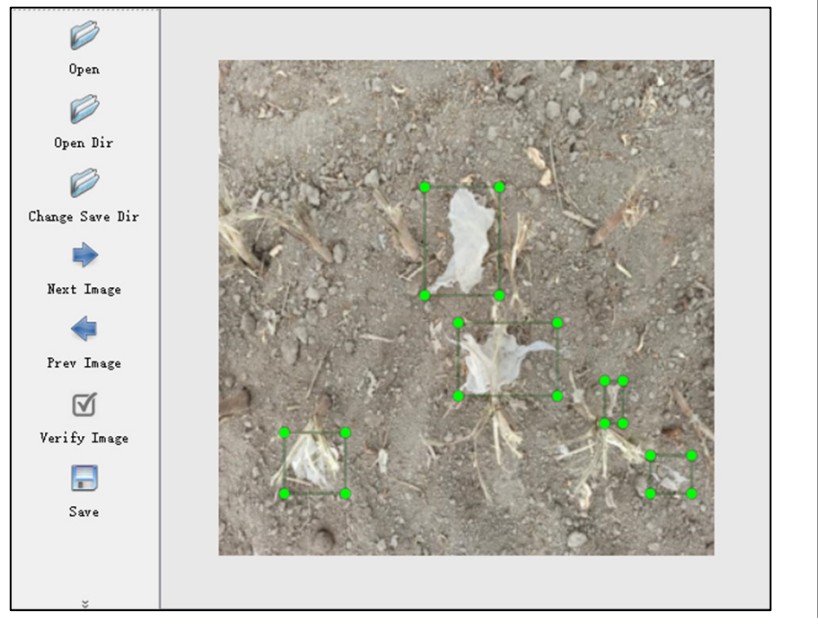

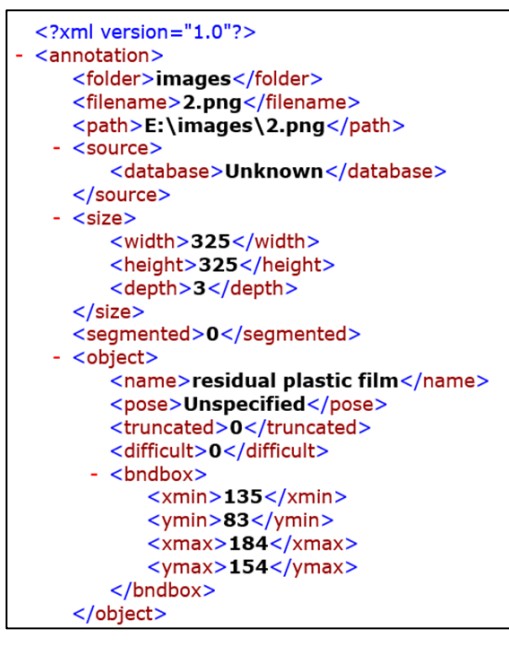

**(a)**    **(b)**

**Figure 3.** Example of residual film annotation. (**a**) Annotation of residual film images; (**b**) information about the XML file.

## 2.2. The Detection Model of Residual Film

Faster R-CNN is a two-stage object detection architecture proposed by Ren, which consists of four stages: input, feature extraction network, region proposal networks, a region with convolutional neural network features, and output [22]. Compared with the one-stage object detection architecture, the two-stage object detection architecture is more complex and slower, but the detection accuracy is higher and more suitable for high-precision, multiscale and small-target problems. After the operation of the residual film recycling machine is complete, the surface will be left with large differences in size, most of which are small-sized residual films. Therefore, Faster R-CNN is adopted as the field surface residual film detection model in this research.

### 2.2.1. Convolutional Block Attention Module

In the captured images of the residual film, only a small portion of the area is occupied by the residual films, while most of the area relates to irrelevant content. To eliminate the featured information that is irrelevant to the residual film, and to improve the feature extraction ability of the residual film detection model, this study introduces the convolutional block attention module (CBAM). The CBAM is introduced in the model to focus on the important features of the residual film and inhibit the unnecessary features.

As shown in Figure 4, the CBAM consists of a channel attention module and a spatial attention module [23]. It assigns a higher weight to the residual film area in the feature map and a lower weight to the background to improve the attention of the neural network to the residual film in the image. The CBAM not only focuses on channel domain information but also focuses on spatial domain information [24]. Compared with SENet, which only focuses

on channel domain information, the CBAM is more suitable for residual film recognition tasks that need to focus on residual film spatial distribution information [25].

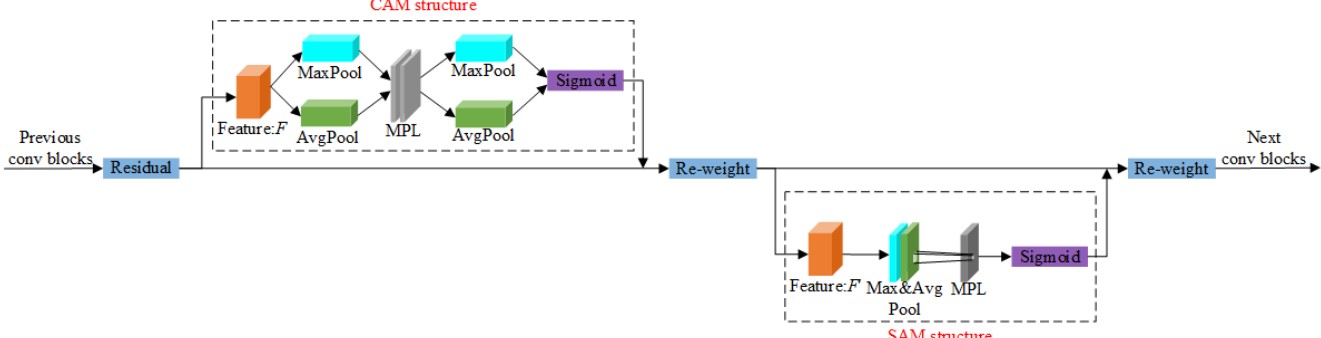

**Figure 4.** CBAM structure diagram.

The workflow of the channel attention module is as follows. Firstly, the input feature *F* is subjected to global maximum pooling and global average pooling operations, and two feature vectors are obtained by the *MLP*. Then, the two feature vectors are used for the add operation and the sigmoid activation operation. Finally, the channel attention feature weight is obtained, which is $M_c$ in Equation (1). The input feature *F* is multiplied by the channel attention weight to obtain the new feature *F′* as the input feature of the spatial attention module, and *F′* is defined in Equation (2).

$$M_C(F) = \sigma(MLP(MaxPool(F) + MLP(AvgPool(F)))) \tag{1}$$

$$F' = M_C(F) \times F \tag{2}$$

where $\sigma$ is the sigmoid activation function, *MaxPool(F)* is the global maximum pooling operation, and *AvgPool(F)* is the global average pooling operation.

The workflow of the spatial attention module is as follows: In the first step, the global maximum pooling and global average pooling operations are performed on the input features *F′* to obtain two feature vectors, and the two feature vectors are stitched together based on the channel. Secondly, a 7 × 7 convolution operation is performed. Finally, the spatial attention feature weight is obtained by sigmoid activation operation, which is *Ms* in Equation (3). The input feature *F′* is multiplied by the channel attention weight to obtain the final feature *F″*, and *F″* is defined in Equation (4).

$$M_S(F') = \sigma(f^{7\times7}([MaxPool(F'); AvgPool(F')])) \tag{3}$$

$$F'' = M_S(F') \times F \tag{4}$$

where $f^{7\times7}$ is a 7 × 7 convolution operation, $\sigma$ is the sigmoid activation function, [*MaxPool(F′)*, and *AvgPool(F′)*] represents the feature splicing operation.

### 2.2.2. Soft Non-Maximum Suppression

The NMS (non-maximal suppression algorithm) is an important part of the object detection algorithm. The calculation formula is shown in Equation (5). Each image generates a large number of redundant prediction boxes in the RPN structure. According to the score matrix and the coordinate information of the prediction box, NMS deletes the prediction box with low confidence and retains the prediction box with high confidence. However, NMS will directly delete the low-confidence prediction box when the residual film overlaps, resulting in missed detection.

$$S_i = \begin{cases} S_i, iou(M, b_i) < N_t \\ 0, iou(M, b_i) \geq N_t \end{cases} \tag{5}$$

where, $N_t$ is the confidence threshold of the set prediction box, $iou(M,b_i)$ is the intersection over the union between prediction boxes, and $M$ is the highest confidence prediction box.

When the IoU (Intersection over Union) value of the prediction box is greater than the set threshold, the soft non-maximum suppression (Soft-NMS) attenuates the confidence instead of completely deleting it. The calculation formula is shown in Equation (6). Using the Soft-NMS, some of the prediction boxes that are suppressed by the NMS but have high confidence may still be retained in subsequent calculations, which improves the accuracy of residual film detection. Thus, this paper replaces the NMS algorithm with the Soft-NMS algorithm.

$$S_i = \begin{cases} S_i & , iou(M,b_i) < N_t \\ S_i(1 - iou(M,b_i)) & , iou(M,b_i) \geq N_t \end{cases} \tag{6}$$

2.2.3. Residual Film Detection Model Based on Multiscale Feature Fusion

In the process of performing object detection, the Faster R-CNN model utilizes deep features to construct feature vectors. After the operation of the residual film recycling machine, the scale of residual films on the surface of the field varies greatly between each other and most of them are small in size. Following multiple convolution and pooling operations, the features of the residual films become fuzzy and are not extracted easily. In addition, the similar color of the residual film and the background interferences make it more challenging for the primitive Faster R-CNN model to extract the fragmented residual film features.

The feature pyramid network (FPN) not only obtains deep-level features but also retains shallow-level features, which is good for solving the problem of multiscale variation in object detection [26]. With the help of the FPN module, multiscale features are integrated to obtain more dependable surface residual film features, ensuring that the feature information of fragmented residual film will not be lost, and to enhance the detection precision of the model.

In this research, we embedded the CBAM module into the FPN structure and designed a network model (MFFM Faster R-CNN) that is more suitable for the detection of residual film on the field surface. It operates the weighting of the residual film feature information at different scales according to channel and space dimensions, which is more beneficial to extract the effective features of the residual film. As shown in Figure 5, the first step is to obtain the residual film feature maps (C2, C3, C4, C5) of each stage by the backbone feature extraction network (in the case of ResNet50). Second, the CBAM module is utilized to give different weights to each residual film feature in channel and spatial dimensions and perform weighting operations, so that the model focuses on the important features of the residual film and inhibits unnecessary features. Then, the number of channels of the residual film feature map in each stage are unified by a series of $1 \times 1 \times 256$ convolution operations. The residual film feature map of the same resolution as the shallow feature map is generated by the upsampling algorithm, which is fused with the shallow feature map to obtain the residual film feature map (B2, B3, B4, B5). In the end, the final residual film feature maps (P2, P3, P4, P5) are obtained by concatenating the $3 \times 3$ convolution operation based on each layer feature map. The residual film feature maps, P2, P3, P4, and P5, are fused with the fused residual film feature map as input to RPN and R-CNN. The Soft-NMS algorithm is used instead of the NMS algorithm in the RPN structure.

*2.3. Experimental Environment and Evaluation Index*

2.3.1. Experimental Environment

This study builds a deep learning framework based on Python 3.7.15 and TensorFlow 2.2.0 and trains the model under the Windows 11 system. The processor is Intel Core i5-12500H (Intel Corporation, Santa Clara, CA, USA), memory is Kingston HX432C18fb 16 GB (Kingston, Taiwan, China), and GPU is NVIDIA RTX3060 (NVIDIA Corporation,

Santa Clara, CA, USA). To improve the training speed of the model, we used GPU to train the model.

**Figure 5.** Structure of the multiscale feature fusion model.

In this study, pretrained weights were used to train the model, which not only improve the convergence speed of the model, but moreover decrease the computational time and cost of the model [27]. To improve the algorithm performance and reduce overfitting, the model parameters were initialized according to Table 1. During training, the momentum was set to 0.937, the initial learning rate was set to 0.001, the learning rate was automatically adjusted using the adaptive moment estimation algorithm, the total number of training rounds was set to 500, the weight decay was set to 0.0005, and the threshold of Soft-NMS was set to 0.3 in the RPN structure. After each training, 10 Epoch saved the network weight file once, saved the parameters of the highest accuracy, and lastly, validated the algorithm with a test set to output the detection results.

**Table 1.** Setting of training parameter value.

| Parameter | Base Learning Rate | Momentum | Iterations | Weight Decay | the Threshold-Value of Soft-NMS |
|---|---|---|---|---|---|
| Value | 0.001 | 0.937 | 500 | 0.0005 | 0.3 |

2.3.2. Evaluation Index

This paper aims to identify and locate the residual film left on the field surface after the operation of the residual film recycling machine. Therefore, it is very important whether the model has missed detection or has made a false detection. To verify the performance of the proposed model, we used four common object detection evaluation indexes: precision, recall, average precision (*AP*), and $F_{1\text{-score}}$.

$$Precision = \frac{TP}{TP + FP} \times 100\% \qquad (7)$$

$$Recall = \frac{TP}{TP + FN} \times 100\% \qquad (8)$$

$$AP = \int_0^1 P(R)dR \qquad (9)$$

$$F_{1-score} = \frac{2PR}{P + R} \qquad (10)$$

where *TP* represents the number of detection boxes that correctly predict the residual film as the residual film, *FP* represents the number of detection boxes that predict other information as residual film, and *FN* represents the number of other information predicted by the residual film.

In addition, in accordance with the physical demand of residual film picking, the model is supposed to focus on the rapid identification of the residual film and determine its position. Accordingly, we also add an important index that reflects the speed of model detection: the average detection time.

$$T_a = \frac{t}{n} \tag{11}$$

where *t* represents the total time spent to detect the test set image, and *n* denotes the number of images in the test set.

## 3. Results and Discussion

### 3.1. Selection of Feature Extraction Network

Faster R-CNN extracts object features by feature extraction network. To obtain deeper feature information about the target, researchers usually deepen the network structure, but as the number of layers of the network structure increases, the problem of network degradation occurs. The problem of network degradation was not effectively solved until the proposal of Residual Network [22]. ResNet maps the redundant network into a shallow network through a jump structure, thereby improving the convergence speed of the model.

The selection of feature extraction network has an important influence on the detection accuracy of the model. To select the feature extraction network suitable for this study, we used, respectively, VGG16, ResNet50, and ResNet101 as feature extraction networks to detect the test set images based on the original Faster R-CNN without any modification [28–30].

The following information can be obtained from Table 2. When VGG16 is used as a feature extraction network, although the average detection speed is the fastest, the *AP* value is the lowest. This is because the VGG16 network structure is relatively simple and the residual film features that can be extracted are limited. When ResNet50 and ResNet101 are used as feature extraction networks, the precision change is more obvious. It is a result of the network structure becoming complex, which can extract more complex residual film feature information, and the false detection rate becomes lower. With the highest *AP* value and the longest average detection time when ResNet101 is used as a feature extraction network. Compared with ResNet101, the *AP* value is reduced by 3.16% when ResNet50 is used as a feature extraction network, while the average detection time is reduced by 65.28 ms. Considering and combining the experimental conditions and the research object of this study, ResNet50 was finally selected as the feature extraction network.

**Table 2.** Detection results of different feature extraction networks.

| Backbone | *Precision* (%) | *Recall* (%) | *AP* (%) | $F_{1\text{-score}}$ | $T_a$ (ms) |
|---|---|---|---|---|---|
| VGG16 | 68.12 | 67.14 | 65.10 | 0.68 | 252.48 |
| ResNet50 | 83.33 | 67.86 | 65.62 | 0.75 | 261.54 |
| ResNet101 | 86.49 | 68.57 | 69.84 | 0.76 | 326.82 |

### 3.2. Ablation Experiment Results

The ablation experiment is mainly used in complex network models, which is an effective method to improve the accuracy of the model by modifying different network structures [31]. To explore the influence of the improved methods on the accuracy of the model, we added FPN, CBAM, and Soft-NMS to the original Faster R-CNN and used the same image dataset for training.

The experiment results are shown in Table 3. Compared with Model-I, the *AP* and $F_{1\text{-score}}$ of Model-II improve by 8.8% and 0.06, respectively. The main reason is that the FPN network combines deep features and shallow features to increase the feature information of the residual film and effectively improve the detection accuracy of the model for small residual films. The average detection time of Model-II is shorter than that of the original Faster R-CNN due to the reduction of the original feature dimension of the model after adding the FPN network, which shortens the detection time.

**Table 3.** Comparison of results of ablation experiments.

| Model | NMS | Soft-NMS | FPN | CBAM | *AP* (%) | $F_{1\text{-score}}$ | $T_a$ (ms) |
|---|---|---|---|---|---|---|---|
| Model-I | √ | | | | 65.62 | 0.75 | 261.54 |
| Model-II | √ | | √ | | 74.42 | 0.81 | 219.68 |
| Model-III | √ | | √ | √ | 78.50 | 0.82 | 237.26 |
| Model-IV (Our model) | | √ | √ | √ | 83.45 | 0.91 | 248.36 |

From the data of Model-III, it can be concluded that when the CBAM module is embedded in the FPN structure, the *AP* and $F_{1\text{-score}}$ of the model are 4.08% and 0.01 higher than those of Model-II, respectively. This is because the CBAM module improves the learning ability of the model for the important features of the residual film and suppresses the irrelevant features. The CBAM module increases the complexity of the model to a certain extent, so the average detection time of Model-III is 17.58 ms longer than that of Model-II.

Model-IV replaces the NMS algorithm with the Soft-NMS algorithm. Compared with Model-III, the *AP* and $F_{1\text{-score}}$ of the entire network model are increased by 4.88% and 0.09, respectively, which can effectively reduce the missed detection rate of the model. The average detection time of Model-IV is increased by 11.1 ms, which is due to the fact that the Soft-NMS algorithm is decaying the confidence level when filtering the prediction boxes instead of directly removing them, thus increasing the model computation time. The model proposed in this study combines the advantages of FPN, CBAM, and Soft-NMS to improve the detection accuracy of residual films and shorten the average detection time.

*3.3. Comparison with Other Object Detection Models*

To verify the detection effect of the model proposed in this paper on the residual film image, two different object detection models, SSD (Single Shot MultiBox Detector) and YOLOv5 (You Only Look Once version 5), were selected to conduct comparison tests [32,33]. The three models were trained under the same experimental environment, experimental index, and residual film dataset.

According to Table 4, the following training results can be obtained. The AP values of SSD and YOLOv5 are lower than the MFFM Faster R-CNN model. As a lightweight object detection model, the SSD model cannot extract complete residual film feature information, so the detection speed is faster, and the detection accuracy is lower. As a single-stage detection model, YOLOv5 does not have an RPN network. It uses the extracted feature information to directly classify the prediction box by regression, and the feature extraction ability of the backbone network cannot be fully utilized. Compared with SSD and YOLOv5, the AP values of MFFM Faster R-CNN increased by 17.02% and 8.64%, respectively, with a significant improvement in accuracy. For the object detection model, the importance of accuracy is the first priority, and the detection time is the second. The average detection time of the MFFM Faster R-CNN model differs from that of SSD and YOLOv5 by no more than 0.1 s, and the model can be subsequently combined with the residual film pickup device. The location of the residual film is identified by the model, and the picking device is controlled by the control unit to pick it up, while the model can be allowed to detect again during the process of the picking device functioning to equalize the excess time.

**Table 4.** Performance comparison of three network models.

| Model | $AP$ (%) | $F_{1\text{-score}}$ | $T_a$ (ms) |
|---|---|---|---|
| YOLOv5 | 66.43 | 0.68 | 192.75 |
| SSD | 74.81 | 0.73 | 176.48 |
| MFFM Faster R-CNN | 83.45 | 0.91 | 248.36 |

The detection results of the same validation set using the three models are shown in Figure 6. It is difficult to determine the specific location of the residual film for SSD, which leads to lower detection accuracy of the model and fails to achieve the expected detection effect. The complexity of the YOLOv5 model is better than that of the SSD model, but in the process of extracting the residual film features, the key feature information of the residual film is not paid any attention, and the correlation between the two dimensions of space and channel is lacking, resulting in poor detection of small residual film. However, our proposed model compensates for the shortcomings of SSD and YOLOv5, which not only ensures high-precision detection but also real-time detection and can effectively detect the residual film on the field surface, which can meet the requirements of automated identification and localization of residual film left on the field surface after the autumn harvest.

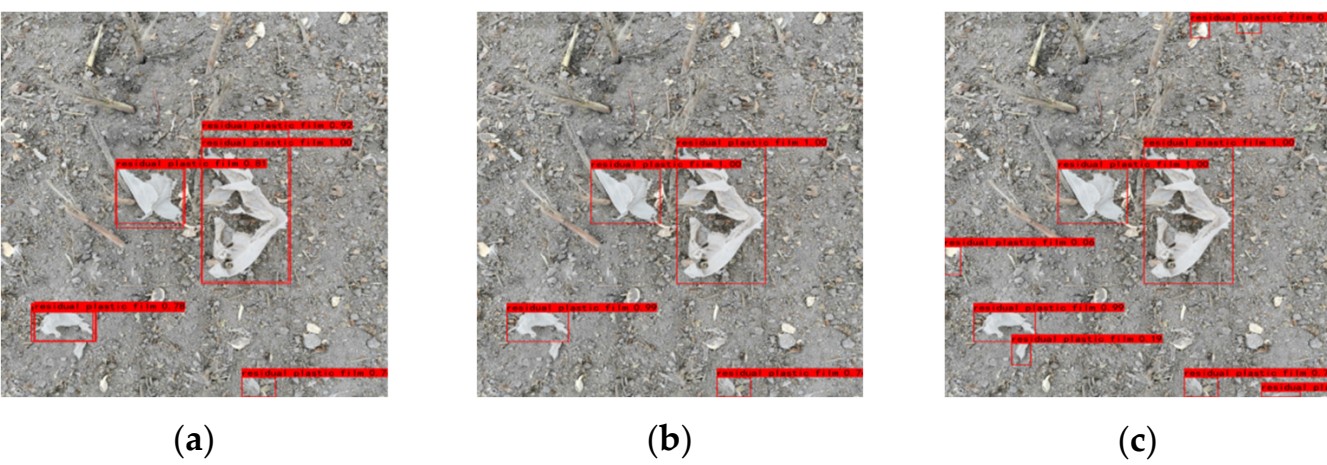

(a)          (b)          (c)

**Figure 6.** Residual film detection results of different models. (**a**) The detection effect of the SSD model; (**b**) the detection effect of YOLOv5; and (**c**) the detection effect of MFFM Faster R-CNN.

### 3.4. APP Development

Graphical user interface (GUI) is the current popular computer desktop graphics application. PyQt5, as a cross-platform GUI library, is widely used by researchers because of its portability and richness [34]. In this study, we designed and developed the residual film identification software based on PyQt5, which allows users to detect residual film by image or video to obtain real-time information on the location of residual film. The user interface of the residual film identification software and the results of the software in detecting residual film in a complex environment are shown in Figure 7.

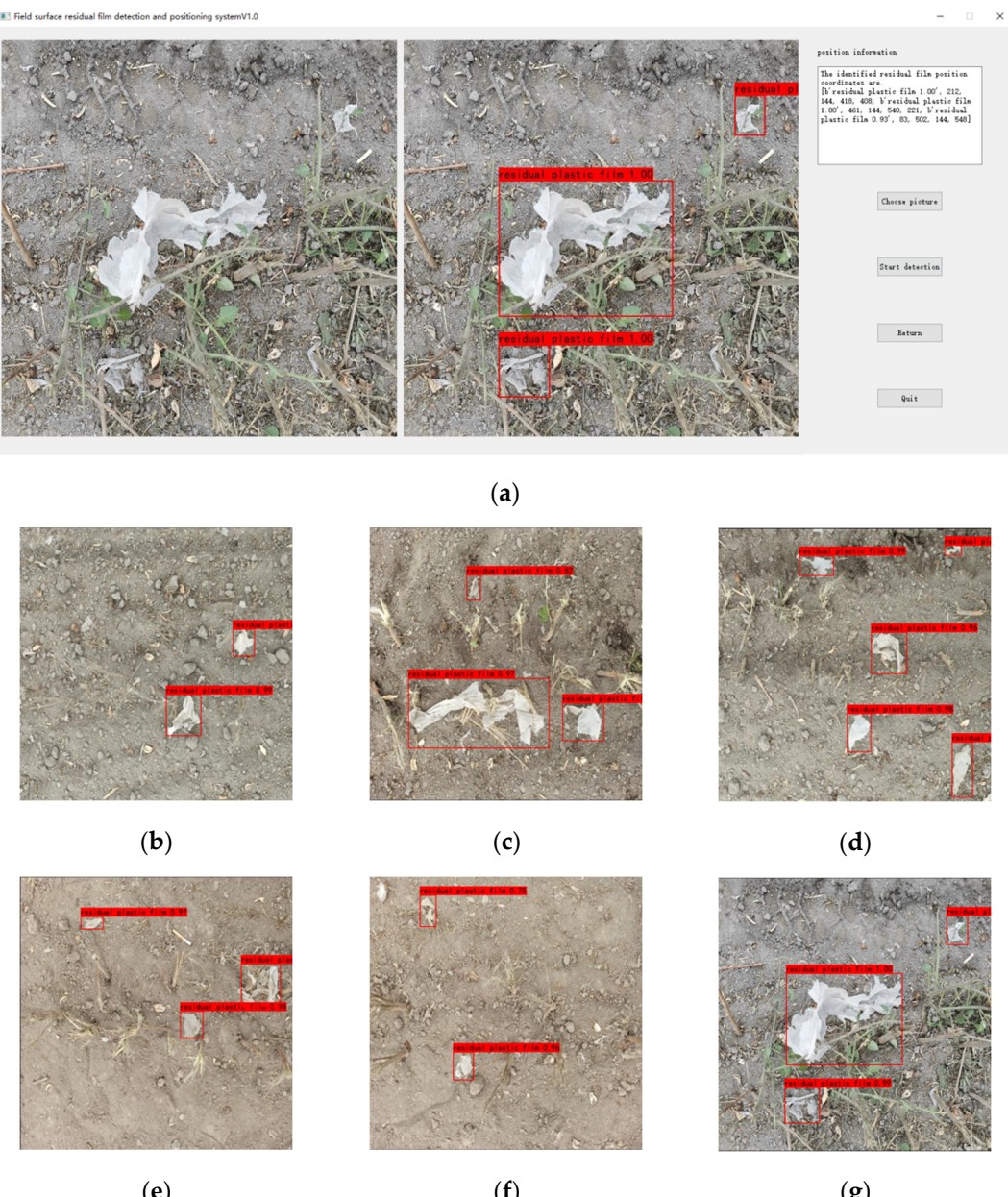

**Figure 7.** Detection results of residual film in complex background. (**a**) Software interface; (**b**) two residual films; (**c**) three residual films; (**d**) more than four residual films; (**e**) small residual films; (**f**) cotton interference; and (**g**) weed interference.

## 4. Conclusions

To achieve accurate detection of residual film on the field surface, this study proposed a model for detecting surface residual film on fields based on the multiscale feature fusion Faster R-CNN model. The main conclusions are as follows:

1.  Based on the Faster R-CNN model, the feature extraction ability of the residual film at different scales was enhanced by using FPN. The residual film regions in the images were focused on by utilizing CBAM, and the missed detection rate of the model was reduced by using the Soft-NMS algorithm instead of the NMS algorithm in the RPN structure. The average accuracy, F1-score, and average detection time of the model were 83.45%, 0.89, and 248.36 ms, respectively.
2.  To investigate the effects of each improvement method based on Faster R-CNN on the model detection accuracy, four sets of ablation experiments were conducted on

a homemade residual film image dataset. The experimental results showed that the AP and F1-score of the MFFM Faster R-CNN model are both higher than the original Faster R-CNN, and the detection speed is also better.

3. To evaluate the detection performance of the proposed model in this study, SSD and YOLOv5 were used to conduct comparison tests. The experimental results showed that the AP value of MFFM Faster R-CNN increased by 17.02% and 8.64% compared with SSD and YOLOv5, respectively, and could identify the residual membrane more effectively. However, compared with SSD and YOLOv5, the average detection time of MFFM Faster R-CNN increased.

4. The MFFM Faster R-CNN model proposed in this study fuses multiscale residual film features, which improves the detection accuracy but does not reduce the detection time. The subsequent research will continue to optimize the detection model and focus on reducing the detection time on the basis of improving the detection accuracy of the residual film. Simultaneously, embedded development will be implemented to apply the detection model to the residual film intelligent picking device; to control the picking component to pick up the residual film.

**Author Contributions:** T.Z.: Conceptualization, methodology, software, visualization, writing of the original draft; Y.J.: funding acquisition and data curation; X.W.: funding acquisition and investigation; J.X.: funding acquisition, formal analysis, writing, review, and editing; C.W.: writing—review and editing, supervision; Q.S.: data curation, investigation, formal analysis, and software; Y.Z.: data curation and investigation. All authors have read and agreed to the published version of the manuscript.

**Funding:** This work was supported by the Xinjiang Agricultural Machinery R&D, Manufacture, Promotion, and Application Integration project (YTHSD2022-09 by J.X.). The authors are grateful for the experimental conditions provided by Xinjiang Agricultural University, the Research Institute of Agricultural Mechanization, and the Xinjiang Academy of Agricultural Sciences.

**Institutional Review Board Statement:** Not applicable.

**Data Availability Statement:** All relevant data presented in the article are stored according to institutional requirements and, as such, are not available online. However, all data used in this manuscript can be made available upon request to the authors.

**Conflicts of Interest:** The authors declare no conflict of interest.

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
