# Peer review of "Detection of Residual Film on the Field Surface Based on Faster R-CNN Multiscale Feature Fusion"

_agriculture, doi:10.3390/agriculture13061158_

Round 1

Reviewer 1 Report

This paper presents an interesting approach for residual film detection in agriculture.

The authors should more clearly articulate their research question and objectives. It would be helpful if they could state in the introduction what they specifically aim to achieve through their proposed residual film detection model.

I would recommend the authors to include a related works section in their paper. This section would serve as a literature review and provide a summary of relevant research that has been conducted in the field, highlighting the similarities and differences between the authors' work and previous research.

The "Materials and Methods" section of this research article provides a detailed description of the dataset and detection model used for residual film detection on field surfaces. The authors constructed a dataset of 1000 residual film images captured with a Sony camera, and expanded it to 3000 images using image enhancement techniques such as center cropping, rotating, and adjusting brightness. 

I recommend providing more details about the dataset: While the authors provide some information about the dataset, such as the number of images and the training, test, and validation sets, they could provide more details. For example, they could describe how the images were captured, the resolution of the images, and any other relevant information about the dataset.

The authors describe the model architecture and the use of the CBAM module, but they could provide more details about the model training process, ideally in a table for better readability. For example, they could describe the hyperparameters used during training, the number of epochs, and any other relevant details about the training process.

The section is well-written and provides clear information about the experiments conducted to improve the accuracy of residual film detection. The authors' approach is well-structured and easy to follow, and the results are presented in a well-organized table format.

While the authors provide detailed information about the different models tested in the study, they do not discuss the limitations of their approach or potential areas for future research. Adding these insights could enhance the value of the study.

Also, the authors should improve the quality of their figures by increasing the resolution and ensuring that all labels and captions are legible.

Reviewer 2 Report

Authors have evaluated faster R-CNN for multiscale feature fusion to detect residual film left in the field after the operation of the residual film recovery machine. The results indicated that the detection is adequately accurate and outperform other models such as SSD and YOLO5. The question left is that the images are collected by a digital camera manually, so how this approach can be implemented over a relatively large field, and also how much the results of the study can be referred and expanded if any other imaging tool is used, such as UAV. 

Also, why is YOLO5 compared not YOLO7 or 8?

The abstract is poorly written. Some sentence is quite confusing, such as, but not limited, "proposed to 15 improve the detection accuracy of residual film by taking the residual film left on the surface after the operation of the residual film recovery machine as the research object"

Also, what are NMS, RPN, AP, F1, SSD, and YOLO there? Please specify each!

Round 2

Reviewer 1 Report

I would like to provide my feedback on the second revision. I am pleased to inform you that the authors have made significant efforts to address the comments and suggestions raised during the initial review. They have implemented almost all of the modifications requested, and as a result, I believe the paper is now suitable for publication.

Reviewer 2 Report

My concerns have been well addressed. Thanks!